# Work Ability and Quality of Life in Patients with Rheumatoid Arthritis

**DOI:** 10.3390/ijerph192013260

**Published:** 2022-10-14

**Authors:** Wojciech Tański, Krzysztof Dudek, Tomasz Adamowski

**Affiliations:** 1Department of Internal Medicine, 4th Military Teaching Hospital, R. Weigla 5, 50-981 Wrocław, Poland; 2Department of Transport Systems, Faculty of Mechanical Engineering, University of Technology, 50-370 Wrocław, Poland; 3Department of Nursing and Obstetrics, Wroclaw Medical University, 51-618 Wrocław, Poland

**Keywords:** quality of life, rheumatoid arthritis, fatigue, work

## Abstract

Background. Reduced work participation has social implications (sickness absence, economic impact) and consequences for the individual patient (impoverishment, depression, limited social interaction). As patients with rheumatoid arthritis (RA) are more likely to experience job loss and/or at-work productivity loss and are at higher risk of sickness absence and, ultimately, permanent work productivity, consideration should be given to the association between work productivity or partial work capacity and quality of life (QoL). The aim of the study was to assess the relationship between QoL and the risk of work disability, as well as to estimate the risk of a future event and identify factors affecting the risk of work disability in RA inpatients. Material and methods. This cross-sectional study included 142 inpatients (65 male) aged 47 (38–58) years, who met the established criteria for a diagnosis of RA and treatment with biologic drugs. Only standardized tools were used to examine the patients: WHOQOL-BREF, MFIS and AS-WIS. Results. An analysis of the QoL scores on the WHOQOL-BREF demonstrated that the patients’ QoL was lowest in the physical health domain and highest in the social relationships domain. The median WHOQOL-BREF total score in the group studied was 62.8, which indicates a moderate QoL. The median total score for the risk of work disability (AS-WIS) was 10.1, which indicates that the level of risk of work disability in the patients was higher than the average level reported in the literature. A multivariate analysis showed that the following were significant independent determinants of a higher risk of work disability: low QoL in the WHOQOL-BREF physical health (β = 0.961; *p* = 0.029) and psychological health (β = 1.752; *p* = 0.002) domains, being in a relationship (β = 0.043; *p* = 0.005) and the use of opioids for pain (β = 3.054; *p* = 0.012). Conclusions. RA patients presented with moderate QoL, moderate fatigue (MFIS) and high risk of disability (AS-WIS). There is an association between a high risk of work disability and lower QoL, especially in the physical and psychological health domains. The lower the QoL in those domains, the higher the risk of work disability. The identification of factors increasing the risk of work disability will help in planning tailored interventions to improve at-work productivity loss and thus prevent work disability.

## 1. Background

Rheumatoid arthritis (RA) affects 1% of the general population [1]. Approximately 400,000 people in Poland suffer from RA [2], and one-third of RA cases are patients over the age of 60 [3]. Rheumatoid arthritis (RA) is a chronic inflammatory disease that significantly affects patients’ daily functioning. An important outcome in working-age RA patients is work disability. Many people with RA stop working very early in the course of the disease, often before being referred to hospital or starting on disease-modifying anti-rheumatic drugs. A significant proportion of RA patients have difficulty with regular engagement in professional activities due to exacerbation of the disease and rapidly progressing disability. It is believed that there may be an association between work participation and disease progression. Difficulties in maintaining a job due to health limitations and the inability to work full time can constitute major problems for patients with RA. According to the literature, the rates of work disability among RA patients range from 22% to 85% in the USA and from 23% to 80% in European countries [4].

Reduced work participation has social implications (sickness absence, economic impact) and consequences for the individual patient (impoverishment, depression, limited social interaction). In the UK, the financial consequences of work disability run into the hundreds of millions of pounds [5]. 

Fatigue is one of the symptoms of rheumatoid arthritis. It is noted in more than 70 percent of patients [6]. Although it is a subjective symptom that also depends on many factors unrelated to the disease, it has a significant impact on the patient’s well-being and functioning, and is associated. Fatigue is also a strong predictor of future disability retirement [7]. About one-third of absenteeism from work is due to musculoskeletal conditions, the most common of which are various forms of arthritis and rheumatic disease [8]. Fatigue is a pervasive symptom in employed people with rheumatic disease. The physical demands of work and fitting into a work schedule were the two reported areas of greatest difficulty, with higher levels of fatigue indicating greater difficulty. Younger workers were more likely to experience fatigue-related difficulties [8]. 

Studies on the subject highlight the role of disease activity and duration, pain and psychological factors related to the disease as predictors significantly associated with permanent work disability in patients with RA [9,10,11,12]. In the case of RA patients, work disability most often results from swelling and damage to joints, which leads to physical limitations and mobility restrictions, whereas early work disability mainly results from inflammation in joints [5]. There is an ongoing discussion in the literature on the predictors with the greatest influence on the ability of RA patients to continue working. Apart from clinical factors, the most commonly reported ones are older age, poor education and performing a job that requires manual labour [13].

In the case of patients with RA, remaining in employment is very important, as work participation is associated with a lower progression rate of RA [14]. It is believed that work disability can stimulate disease progression due to the loss of psychosocial, financial and medical benefits. Studies on this subject highlight the importance of biologic therapies in patients with RA. Early treatment with biologic agents has been associated with lower sickness absence. The Swefot study showed a significant clinical advantage for anti-TNF after the first [15], but not after the second year; the latter finding can be attributed to the the slow, incremental benefit of conventional drugs [16]. However, the two-year results confirmed the advantage of anti-TNF combinations in preventing radiographic damage. The NEO-RACo trial showed that the long-term efficacy of the combination of conventional DMARDs and prednisolone did not improve further when infliximab was added during the first six months of treatment [17]. Moreover, it may delay work disability. However, the amount of research on the subject leaves something to be desired, especially in Poland. In recent years, there has been a shift towards earlier and more aggressive treatment of RA, the aim of which is to prevent and limit the development of erosive joint changes [18]. According to the scientific data, the main problems faced by RA patients include low quality of life (QoL) due to physical and work disability, as well as high costs associated with treatment and the inability to earn an income [19]. According to the literature, one-third of the total costs for RA patients results from production losses, with reduced performance while working (i.e., loss of at-work productivity) and wage loss from quitting or changing jobs, decreased working hours and sick leave having the greatest impact on costs for RA patients [20]. While the Outcome Measures in Rheumatology (OMERACT) initiative has demonstrated a significant loss of at-work productivity in RA patients, no factors associated with at-work performance have been identified. Knowledge of factors associated with at-work productivity loss is necessary in order to prevent both that loss and permanent work cessation due to RA [20,21].

Employment status is an important indicator of pain-related disability. Lower employment rates among patients with chronic pain are associated with higher pain scores and depressive symptoms [22]. Mental health conditions are also significantly related to employment status. Chronic pain patients with mental disorders were more likely to be unable to work due to their condition than those without mental disorders. A significant interaction was found between chronic pain and the outcome of not working in the past year and the number of work days missed per month due to health [23]. However, few studies have examined functional disability and employment outcomes among patients on long-term opioid therapy, despite the fact that patients returning to their regular activities is the main reason for prescribing long-term opioid therapy [24].

Work is an important part of life and is usually the main source of income, directly enabling the satisfaction of material and spiritual needs. Decreased work activity is also due to the coexistence of mental illness, the family situation and the impact of the institution of work and health itself. Prolonging employment participation reflects a significant contribution to the fulfilment of social roles [9]. Identifying the factors associated with work disability in RA patients will make it possible to conduct further research and implement measures aimed at minimising the impact of those factors on QoL. 

As patients with RA are more likely to experience job loss and/or at-work productivity loss and are at higher risk of sickness absence and, ultimately, permanent work disability, consideration should be given to the association between work disability or partial work capacity and QoL. 

Our study had the following research aims:to assess the level of the risk of work disability in RA patients, using the AS-WIS questionnaire;to assess the relationship between QoL and the risk of work disability, as well as to estimate the risk of a future event and identify factors affecting the risk of work disability in RA patients;to determine whether the level of fatigue in patients with RA has a significant influence on their ability to work.

## 2. Study Design and Setting 

The data for the study were obtained from a cross-sectional observational study involving consultations with patients with RA undertaken between September 2020 and December 2021.

### 2.1. Sample

The patients who were invited to take part in the study (*n* = 183) had been treated at the hospital clinic and attended follow-up appointments as part of their biologic treatment. Not all the patients completed the study questionnaires. Thus, a total of 142 patients were ultimately included in the analysis. The remaining 41 patients did not meet all the study criteria (mainly as regards work status) or did not fill out all the study documentation.

### 2.2. Procedures

The patients included in the study were examined by a team consisting of a rheumatologist and an internal medicine nurse who had been trained in how to carry out the tasks within the study. The physician evaluated the patients’ clinical data. The role of the nurse was to provide assistance to the patients when they had questions regarding the questionnaire or the study. The main objective was to obtain questionnaires that the patients had completed themselves. Clinical data were obtained from the patients’ medical records. 

The patients were informed about the purpose and procedure of the study and were advised that they could withdraw from it at any time. 

The study was approved by the relevant Bioethics Committee, at the Military Medical Institute in Warsaw (approval no. 170/2020; date: 10 May 2020). All procedures followed were in accordance with the ethical standards of the responsible committee on human experimentation (institutional and national) and with the Helsinki Declaration of 1975, as revised in 2000. Written informed consent was obtained from all individual participants included in the study.

### 2.3. Inclusion and Exclusion Criteria

Respondents of the screening questionnaire had to meet the following inclusion criteria: age 18–70 years, diagnosis of RA, treatment with a biologic in accordance with the guidelines by the American Rheumatism Association, being in paid employment, voluntary provision of informed consent to participate in the study and ability to complete the study questionnaires independently. Patients were excluded from the study when they met even one exclusion criterion: age <18 years, diagnosed mental disorders treated pharmacologically (depression or mental conditions involving low mood), no possibility to fill out the study questionnaires independently, coexistence of other chronic conditions that may interfere with the main results and QoL (cancer, renal failure, NYHA class IV heart failure).

### 2.4. Instruments

World Health Organization Quality of Life-BREF (WHOQOL-BREF)—a generic questionnaire consisting of 26 items, which measures four domains, i.e., physical health, psychological health, social relationships and environment, as well as overall QOL and general health in the past 14 days. Domain items are summed and transformed into a 0–100 score. The higher the score, the better the patient’s health-related quality of life (HRQoL). The WHOQOL-BREF has good internal consistency, sensitivity to change and discriminant validity, which means that it demonstrates excellent ability to discriminate between the ill and well groups [25].

All items satisfied the generally adopted criteria. In the group of 142 subjects, Cronbach’s alpha (α) was 0.929, and the mean correlation between items I was 0.341. The mean score (M) was 86.9 and the standard deviation (SD) was 13.9.

2.Modified Fatigue Impact Scale (MFIS)—comprises 21 items in three scales (physical, cognitive and psychosocial functioning), which can be combined into a total score. Each item is scored between 0 and 4. Thus, the questionnaire has a possible total score of 84. The higher the score, the greater the impact of fatigue on QoL [26]. An analysis of the reliability of the MFIS in a group of 142 patients with RA confirmed that the scale has satisfactory psychometric properties. All of its items satisfied the generally adopted criteria. Cronbach’s alpha (α) was 0.948, and the mean inter-item correlation I was 0.517. The mean score (M) was 45.7 and standard deviation (SD) was 16.7.3.Ankylosing Spondylitis Work Instability Scale (AS-WIS); the instrument was used in the study with the consent of the University of Leeds. The AS-WIS is a self-administered questionnaire that enables the risk of work disability to be monitored. It consists of 20 statements; answers are given using a TRUE/NOT TRUE format depending on whether the statements apply to the respondent. It is a simple, validated screening instrument for work instability (the consequences of a mismatch between a patient’s functional ability and their job tasks). It enables health professionals to monitor the risk of work disability in patients with ankylosing spondylitis [27]. As there are no questionnaires designed to measure work instability specifically in patients with RA, an analysis of the psychometric properties of the AS-WIS was carried out for the purposes of the present study. The results of the psychometric analysis of the tool in a group of 142 RA patients and the analysis of the reliability of the items of the original 20-item scale (Table 1) showed that the mean inter-item correlatiI (*r*) was 0.315, which is lower than the lowest acceptable value of 0.4 (Klein’s criterion). The mean score in the group of 142 patients was 13.6 (SD = 5.3), the standardised Cronbach’s alpha was 0.869 and the mean inter-item correlIon (r) was 0.315. Items 4, 5, 9 and 18 showed a weaker correlation with the total score compared with other items (Figure 1). After these four items were removed, the psychometric properties of the scale improved: the mean score was 10.1 (SD = 5.1), the standardised Cronbach’s alpha was 0.921 and the mean inter-item corIation (r) was 0.430. The 16-item scale was found to be reliable: particular items were correlated with the total score at a level of least 0.4 (Klein’s criterion) and the Cronbach’s alpha was higher than 0.7 (Nunnally’s criterion). Therefore, the Polish 16-item version of the AS-WIS was used in the RA patients studied.

As expected, there was a significant correlation between the work disability risk scores on the two scales (*r* = 0.989, *p* < 0.001, Figure 1). For the shortened 16-item version of the AS-WIS, the cut off differentiating between RA patients at high risk of work disability and those at low or moderate risk of work disability was a score of 14 or higher (Figure 2).

### 2.5. Statistical Analyses

Statistical analyses were performed using the STATISTICA v. 13.3 software (TIBCO Software Inc., Palo Alto, CA, USA). For quantitative variables, median (Me), lower quartile (Q1) and upper quartile (Q3) were calculated. Empirical distribution fit to a normal distribution for quantitative variables was verified using the Kolmogorov–Smirnov and Shapiro–Wilk tests. Spearman’s rank correlation coefficient was calculated to assess the correlation between monotonic relationships between variables. Qualitative (nominal and categorical) variables were reported in contingency tables as numbers (*n*) and percentages (%). Continuous variables were converted into dichotomous variables using cut-off values determined by ROC curve analysis. The significance of differences in quantitative parameters between two groups was verified using the Mann–Whitney U test, and the independence of two qualitative parameters was verified using Pearson’s Chi squared test. The reliability of the items in the AS WIS original scale consisting of 20 items and shortened to 16 items was assessed by calculating the Cronbach’s alpha and the mean correlation between items r. Multivariate logistic regression analysis was used to establish independent predictors of high risk of disability. The goodness of fitting the model to the observed results was assessed using the Hosmer–Lemeshow test. For all statistical tests, a significance threshold of *p* < 0.05 was used.

## 3. Results

### 3.1. Analysis of QoL, Fatigue Levels and Risk of Work Disability in 142 Patients with RA

An analysis of the QoL scores on the WHOQOL-BREF demonstrated that the patients’ QoL was lowest in the physical health domain (43) and highest in the social relationships domain (75) (Table 2). The median WHOQOL-BREF total score in the group studied was 62.8, which indicates moderate QoL. Similarly, the median total fatigue score on the MFIS in the group studied was 45, which indicates a moderate level of fatigue. The level of fatigue was highest for the psychosocial functioning (6) and physical functioning subscales (21). The median total score for the risk of work disability (AS-WIS) was 10.1, which indicates that the level of risk of work disability in the patients was higher than the average level reported in the literature. 

### 3.2. Spearman’s Rank Correlation Analysis between QoL and the Risk of Work Disability 

A statistically significant negative correlation was observed between the domains of QoL and the risk of work disability and the level of fatigue (Table 3). The strongest association was noted for the physical and psychological health domains, overall QoL and the WHOQOL-BREF total score. The weakest association was found for the social relationships and environment domains and for general health.

We carried out further analyses in two groups of patients defined according to the level of risk of work disability based on the results of the AS-WIS questionnaire: group 1: low-to-moderate risk of work disability—*n* = 97group 2: high risk of work disability—*n* = 45

An analysis of the sociodemographic and clinical characteristics of the patients studied, who were divided into two groups depending on the level of risk of work disability, showed statistically significant differences between the two groups in terms of marital status. Patients at low or moderate risk of work disability were significantly more likely to be single compared with patients at high risk of work disability (25.8% vs. 6.7% (*p* = 0.007)) (Table 4). No statistically significant differences were seen as regards the other variables. The median age of the patients studied was 47 years. Most patients had higher education qualifications (52.1%). More than half of the patients studied were male (54.2%) and 77.5% were in full-time jobs. Thirty-five per cent of the patients had comorbidities, the most common being hypertension (17.6%), hypothyroidism and diabetes (7.1%).

A comparative analysis between the two groups in terms of pain-relieving and other medications used for RA before the initiation of biologic therapy showed no statistically significant differences (Table 5), with the exception of opioids. Patients at low risk of work disability were significantly less likely to have been using opioids before the initiation of biologic therapy, compared with patients at high risk of work disability (1.0% vs. 8.9% *p* = 0.035) (Table 5). 

A comparative analysis between the two groups in terms of the biologics used showed no differences between them, except for rituximab, which was significantly more commonly used in patients at low risk of work disability compared with patients at high risk of work disability (13.4% vs. 2.2%; *p* = 0.038) (Table 6).

### 3.3. Analysis of QoL and the Level of Fatigue Relative to the Level of Risk of Work Disability (AS-WIS)

A comparative analysis between the groups in terms of the scores on the domains and general items of the WHOQOL-BREF showed that, when compared with patients at high risk of work disability, patients at low risk of work disability had a significantly higher median WHOQOL-BREF total score (65 vs. 58; *p* = 0.003), as well as significantly higher median scores for the physical health domain (47 vs. 32 (*p* < 0.001)) and psychological health domain (71 vs. 63 (*p* = 0.005)) and overall QoL (4 vs. 3 (*p* = 0.006)). No statistically significant differences were found between the groups in terms of the level of fatigue as measured by the MFIS (Table 7).

### 3.4. Analysis of the Relationship between the Selected Predictors and the Risk of Incapacity for Work

The results of the independence tests and estimated values of the odds ratios showed that the risk of incapacity for work was greater in patients who had used analgesic opioids before starting biological therapy (OR = 9.37; *p* = 0.035), patients with low WHOQOL Total score (<72) (OR = 10.1; *p* = 0.001), patients with low QoL in the WHOQOL-BREF physical health (<39) (OR = 3.86; *p* < 0.001), psychological health (<79) (OR = 5.62; <0.001) and social relationships (<75) (OR = 2.33; *p* = 0.020) domains, older patients (>58 years) (OR = 2.49; *p* = 0.017), patients with obesity (BMI > 28.6 kg/m^2^) (OR = 2.49; *p* = 0.020) and less educated patients (OR = 2.05; *p* = 0.049). Patients not treated with rituximab had a higher risk of work disability (OR = 6.81; *p* = 0.0380). A similar finding was made for patients who were in a relationship (OR = 4.86; *p* = 0.15). The results are shown in Table 8.

Multivariate logistic regression analysis showed that significant independent predictors of high risk of work disability are: low quality of life in the WHOQOL-BREF domains, physical health (β = 0.961; *p* = 0.029) and mental health (β = 1.752; *p* = 0.002), in the relationship (β = 0.043; *p* = 0.005) and the use of opioids in the treatment of pain (Table 9). The results are as follows: Logit *p* {*Y* = *High level*|*X*} = −4.36 + 3.05 * {*Opioids = yes*} + 1.75 * {*Domain* 2 <79} + 2.04 * {*Married*} + 0.96 * {*Domain* 1 <39}

Sensitivity = 84.4%, specificity = 63.9%, Acc. = 70.4%, PPV = 52.1, NPV = 89.9, LR(+) = −2.34

Group-Goodness of fit: Hosmer-Lemeshow test = 0.0831, *p*-value = 0.99916.

**Table 9 ijerph-19-13260-t009:** A univariate and multivariate logistic regression analysis of the risk of incapacity for work from selected variables and estimated values of the odds ratios.

Predictors	Univariate		Multivariate		
	b	*p*	b	*p*	OR (95% CI)
Age > 58 years	0.913	0.018	0.735	0.148	2.08 (0.77–5.63)
BMI > 28.6 kg/m^2^	0.913	0.022	0.407	0.410	1.50 (0.57–3.95)
Lack of higher education qualifications	0.717	0.051	0.437	0.367	1.55 (0.60–4.00)
WHOQOL Total <72	2.312	0.002	1.919	0.116	6.61 (0.62–74.4)
WHOQOL Domain 1 <39	1.351	<0.001	0.961	0.029 *	2.62 (1.11–6.19)
WHOQOL Domain 2 <79	1.727	<0.001	1.752	0.002 **	5.77 (1.92–17.3)
WHOQOL Domain 3 <75	0.845	0.022	−0.624	0.602	0.77 (0.28–2.08)
WHOQOL Domain 4 <64	0.730	0.052	0.183	0.728	1.20 (0.43–3.37)
Overall QoL <4	0.906	0.015	−0.361	0.520	0.70 (0.23–2.09)
General health <3	0.561	0.128	−0.346	0.511	0.71 (0.25–1.99)
MFIS Physical subscale >21	0.644	0.081	0.104	0.832	1.11 (0.42–2.91)
In a relationship	1.581	0.014	2.043	0.005 **	7.72 (1.83–32.6)
Rituximab: no	1.918	0.069	−1.006	0.376	0.37 (0.04–3.40)
Opioids before biologic therapy	2.237	0.048	3.054	0.012 *	21.2 (1.97–228)

Domain 1—physical health; Domain 2 psychological health; Domain 3—social relationships; Domain 4—environment; MFIS—Modified Fatigue Impact Scale; OR—odds ratio; CI—confidence interval; *—*p* < 0.05; **—*p* < 0.01.

## 4. Discussion

In the subject literature, the authors stress that moderate RA still has a great impact on the patient’s ability to work [28], with the impact being comparable to that observed in patients with other chronic conditions, such as chronic obstructive pulmonary disease, asthma and irritable bowel syndrome [29]. Our study aimed to determine the extent to which RA limits patients’ ability to work, assess the level of risk of work disability in RA patients and identify factors significantly associated with the risk of work disability. Our findings show that the patients from the hospital biologic treatment clinic who participated in our study had a moderate/high level of risk of work disability. When comparing our findings with those reported in the literature, it can be concluded that the patients included in our study had a higher risk of work disability compared with patients in other studies [28,29,30]. Van Vilsteren et al. showed in their study that RA patients lose approximately four hours of productive work per two weeks, assuming an average work week of 28.7 h [30]. In a study by Galloway et al., over half of the RA patients studied reported that their employment status had changed due to their condition. Moreover, a mean 29% reduction in productivity at work was recorded in those patients with RA who were still employed [28]. According to the authors of the literature on the subject, a reduction in work productivity may result from both external and internal factors. Those employees who have a poorer mental health status and more physical limitations report a greater number of work hours lost compared with patients treated with biologics and even compared with those employees who are not satisfied with their job. In the present study, we did not carry out a separate analysis of the association between the mental condition of the patients studied and the work disability risk level. The causal relationship between quality of life and work disability is bidirectional, especially with regard to psychological health. Physical activity significantly reduces depressive symptoms among people with mental illness [31]. urOur findings with respect to patient QoL in the psychological domain, as measured using the WHOQOL-BREF, are in line with the findings from a study by van Vilsteren et al., who noted that low QoL in the psychological domain had a negative impact on work ability in RA patients [29]. The levels of work disability reported in the literature referred to in our study vary considerably depending on many factors, including, in particular, disease severity and activity, severity of pain and the research tools used. 

The literature on the association between work disability and QoL in RA patients is still limited and no studies on the level of work disability in Polish patients with RA have been reported to date. There are few studies that have comprehensively investigated factors that may be associated with work limitations in RA patients other than those associated with disease progression [4,8]. Patients with disabilities, in addition to functional limitation, have more mental health risks than people without disabilities [32]. This indicates a casuistic link between disability and low QoL. Disability is a consequence of the interaction of a person’s health and individual characteristics with social factors [33]. In our present study, we carried out comparative analyses aimed at identifying such factors. Our findings showed that the groups which were compared, namely a group of patients at high risk of work disability and a group of patients at low risk of work disability, differed in terms of marital status. Patients at high risk of work disability were statistically significantly less likely to be single. This may be explained by the fact that single patients cannot afford to be unable to work, despite physical and psychological dysfunctions, as they have no other sources of income. On the other hand, patients who are in a relationship are more likely to be able to count on support, including financial support, from their partner or family.

Another interesting finding from the present study concerned the association between the medications used for RA and pain and the level of risk of work disability. Our analyses found that patients who had been treated with opioids before they were started on a biologic agent had a significantly higher risk of work disability. Moreover, our findings showed that patients treated with rituximab had a significantly lower risk of work disability. Our findings contribute to the ongoing discussion on the benefits of biologic therapy in patients with RA [34,35].

The availability of biologic drugs over the past ten years has increased expectations of reduced rates of work disability in patients with RA. However, reports based on clinical cohorts have not demonstrated that biologic agents have a major impact on the work status of RA patients [1]. In many situations, the timing of biologic treatment may not be optimal and biologic drugs may be used infrequently for financial reasons [1]. This was also the case with the patients included in our study, who were referred to the hospital biologic treatment clinic because traditional treatment options were contraindicated or ineffective, and who had severe disease. 

Rituximab is used in combination with methotrexate in the treatment of severe RA. It has been shown to reduce joint damage progression and improve physical fitness in RA patients [35]. Studies have shown that biologic therapy has a positive impact on the ability of patients with RA to work and improves their QoL by improving their physical fitness. Our OR analysis showed that those patients who were not treated with rituximab had a higher risk of work disability. However, a lack of treatment with rituximab was not identified as a significant independent determinant of increased risk of work disability in multivariate analysis, which may be due to the advanced disease status in the patients studied. In a study by Tanaka et al., patients treated with tocilizumab had better health scores and showed a greater improvement in work productivity, which translated into a better QoL and better mental health [36]. Similarly, in the study by Rizza et al., simplifying the treatment regimen for patients with diabetes had the expected effect of increasing the quality of life for many elderly patients [37].

It has been stressed in the literature on the subject that improving work productivity is important in improving QoL in RA patients [38]. As expected, one very important factor in such analyses is the duration and stage of disease. Early treatment of RA leads to early remission, low disease activity and a lower level of disability. Moreover, it results in improvement in activity impairment, which in turn leads to less work productivity loss [38]. It has been reported that the long-term administration of tocilizumab in patients with RA facilitates remission. In one study, the authors noted that biologics are licensed for use in patients with severely active or moderate RA who have not responded to standard treatment with disease-modifying antirheumatic drugs [39]. In a study by Tektonidou et al., treatment with adalimumab resulted in improved work productivity and improved sleep problems in patients with moderate to severe RA [40]. The treatment yielded decreases in the percentage of work missed, work impairment while working, overall work impairment and activity impairment from baseline to month 24 [40]. Similar findings were reported from a study by Michaud et al., in which treatment with adalimumab or baricitinib resulted in reductions in both pain and fatigue, as well as improvements in daily activity and work productivity compared with placebo [41]. The authors also found that pain and fatigue tended to be more correlated with daily activity and work productivity compared with disease activity parameters.

An interesting result of the study is that, contrary to literature reports [42,43,44], the association of overlap between RA and diabetes with a higher risk of disability was not confirmed. The overlap between the two conditions can exacerbate disability and significantly lower QoL through the occurrence of cardiovascular events. Patients included in the study did not have their medical history analyzed for a history of cardiovascular events such as stroke or myocardial infarction that would limit fitness and activity. A second reason for the lack of association between diabetes comorbidity and work activity is the young mean age of the patients studied (47 (38–58) years) and the low percentage of patients with diabetes participating in the study (6.3%).

The literature on the subject highlights the significance of pain as a predictor of longer work absence [45,46]. Our present study did not examine the association between pain and work disability. However, our correlation analysis showed that the use of opioids for pain before the initiation of biologic therapy was a predictor of a higher risk of work disability in RA patients. It should be noted that RA patients who are treated with opioids are undoubtedly those with a high severity of RA, which may be associated with significant limitations in physical and mental functioning. Only a small number of patients included in our study had been treated with opioids before they started biologic therapy. However, those patients, too, improved after the initiation of biologic treatment and could discontinue treatment with opioids. Opioids are used in those patients who do not respond to standard treatment. Treatment with opioids may have salutary effects for those patients as it can provide great pain relief and yield a clear improvement in QoL. However, there is evidence that opioids have no beneficial effect compared to other pain-relieving medications [46]. Epidemiological data suggest that patients treated with opioids have a higher risk of respiratory tract infections and higher mortality related to, e.g., drug overdose and cardiovascular causes [46].

In our study, we found a significant negative correlation between the risk of work disability and QoL in terms of all domains and general items of the WHOQOL-BREF. Our findings are consistent with those of other studies. Van Vilsteren et al. found that at-work productivity loss is negatively associated with health-related QoL, especially as regards the domains of mental health, physical role limitations and pain [33]. In our study, a lower QoL in terms of all WHOQOL-BREF domains and general items significantly increased the risk of work disability. Moreover, our multivariate analysis showed that a low QoL in the WHOQOL-BREF physical health and psychological health domains was a significant independent determinant associated with a higher risk of work disability. 

As was the case with our findings, a study by Chorus et al. showed that RA patients experience significant limitations in physical role functioning, including work, and found that work is a significant independent external determinant of physical HRQoL, but not of mental HRQoL [47]. 

Another determinant discussed in the literature on the subject that may be significantly associated with work disability is chronic fatigue [28,48,49]. In the present study, the levels of fatigue in the patients studied were assessed using the MFIS, which measures the levels of physical, cognitive, psychosocial and total fatigue. The patients reported moderate/high levels of fatigue in all domains of the MFIS. Our comparative analyses showed that patients at high risk of work disability had a higher median total MFIS score. However, the difference was not statistically significant. Interestingly, fatigue was not shown to be a significant determinant affecting the risk of work disability in our multivariate analysis. In the few available studies, fatigue levels were assessed using different tools, which may make it difficult to interpret their results. The results of those studies in which the level of fatigue was assessed using the same questionnaire vary. It has been shown that there is an association between the level of fatigue and Disease Activity Score (DAS-28) [48]. The van Amelsvoort et al., study found that fatigue is a strong predictor of future disability retirement [7]. Galloway et al. have shown in their study that there is a significant association between the level of fatigue and the treatment used, notably corticosteroids. In our present study, glucocorticoids were more likely to be used by patients at high risk of work disability. However, as has been noted in other studies, the rate of use of glucocorticoids in patients with RA is not high. In our present study, 12% of the patients studied were treated with glucocorticoids [28]. It may be necessary to use a different tool for the assessment of fatigue in further studies on the association between the level of fatigue and disease activity status and the level of work disability. When planning analyses, it is worth taking into account the baseline health status of patients as there is evidence indicating an association with better outcomes in long-term studies.

The present study highlights one of the most important economic problems, namely work disability in patients with chronic conditions. Moreover, it emphasises the significance of QoL as a predictor of work disability. It is worth paying more attention to other issues relating to RA than the medical aspects of the disease.

## 5. Limitations of the Study

The first and main limitation of our study is that we assessed the risk of work disability in the patients studied using the AS-WIS, which is not specific to RA. Furthermore, due to the lack of uniform assessment standards in rheumatology, it is difficult to compare our findings with those of other studies. When assessing the level of risk of work disability in RA patients, it is worth taking into consideration the type of work the patients do, as it may have a significant impact on study results. Moreover, QoL should be studied in the context of a prospective study, in which QoL may worsen or improve over time with treatment outcome. The study should include, for example, a patient diary or reassessment of QoL. Another limitation of the study is the lack of information on the baseline work capacity of the patients studied. One interesting complement to our study would be an assessment of the dynamic of change in work ability over the duration of biologic therapy, which was not undertaken in the present study. Moreover, when recruiting patients, we did not set any limitations as to the duration of disease. Thus, some patients had a long disease duration and might not have fully remembered their work history. Another limitation is the size of the study group, as well as the fact that it is a single-centre study that mainly included patients attending follow-up appointments at the biologic therapy clinic, who may not be representative of the entire population of RA patients. 

## 6. Practical Implications

It is necessary to standardise diagnostic practices and implement tools that would complement clinical assessment and thus enable the determination of the range of personal factors involved as well as the scale of the problem of work disability in patients with RA, including the associated physical, psychological and economic consequences. This means that physicians should not only focus on improving disease severity when treating RA patients who are struggling to maintain their work productivity, but they should also focus on personal and work-related factors to ensure a more holistic approach. 

It is appropriate to consider delivering educational activities for health care workers to improve the medical management of patients with RA. Further studies focusing on the assessment of work ability in RA patients and identifying the most vulnerable patients are necessary. The identification of factors increasing the risk of work disability will help plan tailored interventions to improve at-work productivity loss and thus prevent work disability.

## 7. Conclusions

RA patients presented with moderate QoL, moderate fatigue (MFIS) and high risk of disability (AS-WIS). There is an association between a high risk of work disability and lower QoL, especially in the physical and psychological health domains. The lower the QoL in those domains, the higher the risk of work disability. The use of opioids for pain before the initiation of biologic therapy is a statistically significant independent determinant of higher risk of work disability, even where opioids can be discontinued as a result of biologic treatment. Being in a relationship is a statistically significant independent determinant associated with a higher risk of work disability. While our findings should be interpreted with caution, they provide insight into which RA patients are at risk of at-work productivity loss. 

## Figures and Tables

**Figure 1 ijerph-19-13260-f001:**
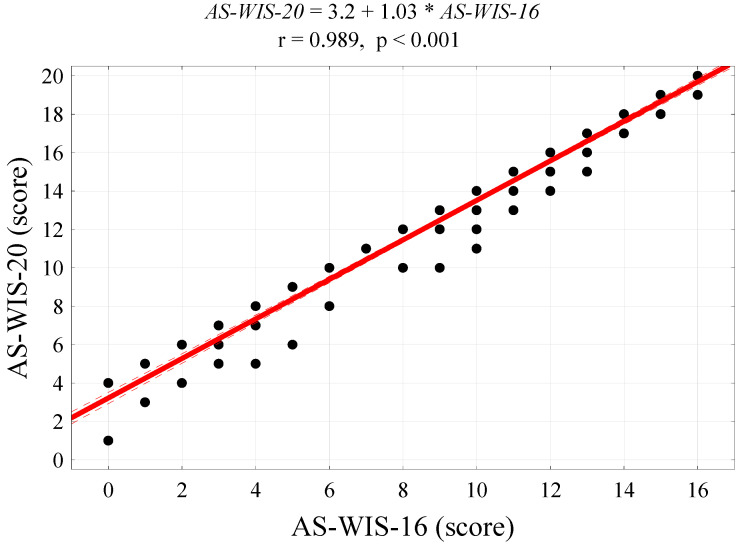
A correlation diagram between work disability risk scores on the original scale (AS-WIS-20) and work disability risk scores on the shortened version of the scale (AS-WIS-‘6), Pearson’s correlation coefficient and regression line.

**Figure 2 ijerph-19-13260-f002:**
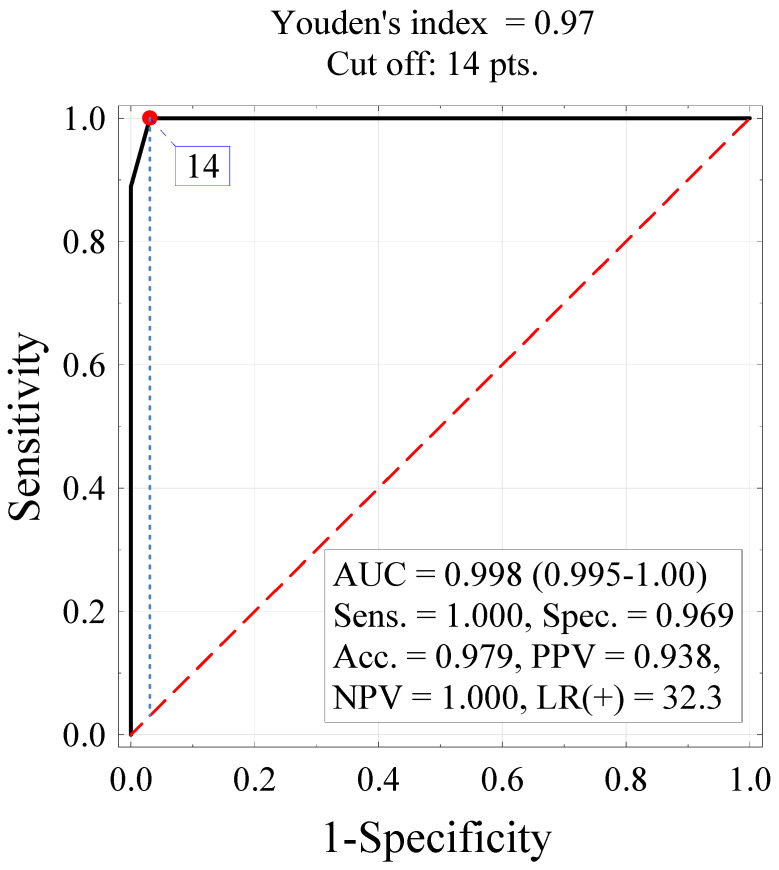
An ROC curve for identifying a high risk of work disability based on the score on the shortened version of the AS-WIS (AS-WIS-16). AUC—area under the curve, Sens.—sensitivity, Spec.—specificity, Acc.—accuracy, PPV—positive predictive value, NPV—negative predictive value, LR(+)—likelihood ratio.

**Table 1 ijerph-19-13260-t001:** An analysis of the reliability of the AS-WIS items and of the scale’s shortened version in a group of 142 patients with RA.

Item	M *	SD *	r *	α *	M *	SD *	r *	α *
1	12.9	5.0	0.621	0.899	9.3	4.7	0.638	0.917
2	13.1	4.9	0.641	0.898	9.6	4.7	0.672	0.916
3	12.9	4.9	0.664	0.897	9.3	4.7	0.673	0.916
4	12.7	5.1	0.310	0.906	-	-	-	-
5	12.8	5.2	0.096	0.911	-	-	-	-
6	12.8	5.0	0.546	0.901	9.2	4.8	0.533	0.920
7	13.0	4.9	0.713	0.896	9.4	4.7	0.736	0.914
8	12.8	5.0	0.469	0.902	9.3	4.9	0.400	0.923
9	12.6	5.2	0.084	0.908	-	-	-	-
10	13.0	4.9	0.689	0.896	9.5	4.7	0.704	0.915
11	13.1	4.9	0.742	0.895	9.6	4.7	0.748	0.914
12	13.0	5.0	0.462	0.903	9.5	4.8	0.502	0.921
13	13.1	4.9	0.613	0.899	9.6	4.7	0.615	0.918
14	13.1	4.9	0.593	0.899	9.5	4.7	0.606	0.918
15	13.2	4.9	0.652	0.898	9.7	4.7	0.655	0.916
16	12.9	4.9	0.664	0.897	9.4	4.7	0.661	0.916
17	12.7	5.1	0.475	0.902	9.2	4.9	0.457	0.921
18	12.7	5.2	0.108	0.909	-	-	-	-
19	13.0	4.9	0.638	0.898	9.4	4.7	0.649	0.917
20	12.9	4.9	0.712	0.896	9.4	4.7	0.728	0.914
	M = 13.6, SD = 5.3, α = 0.869, r = 0.315				M = 10.1, SD = 5.1, α = 0.921, r = 0.430			

M *—mean total score after item removal, SD *—standard deviation of the total score after item removal, r *—correlation between a given item and the total score (without a given item), α *—internal consistency of the scale (alpha coefficient) if the item concerned were to be removed.

**Table 2 ijerph-19-13260-t002:** The QoL, level of fatigue and risk of work disability in the patients studied.

Characteristics	Me	Q1–Q3
Domain 1. Physica-health (0–100 score)	43	32–54
Domain 2. Psychologica-health (0–100 score)	71	54–83
Domain 3. Social relat-onships (0–100 score)	75	58–75
Domain 4. Env-ronment (0–100 score)	61	74
WHOQOL-BR-F Total (0–100 score)	62.8	53.3–71.0
Physical subscale (0 to 36)	21	15–26
Cognitive subscale (0 to 40)	19	13–25
Psychosocial subscale (0 to 8)	6	4–8
Total MFIS score (0 to 84)	45	33–56
Total AS-WIS score	10	5.1

MFIS—Modified Fatigue Impact Sc–le; AS-WIS—work disability risk score, Me––median, Q1—lower q–artile; Q3—upper quartile.

**Table 3 ijerph-19-13260-t003:** Spearman’s rank correlation coefficients (Rho) between QoL scores (WHOQOL-BREF) and the risk of work disability (AS-WIS) and the level of fatigue (MFIS) in a group of 142 patients with RA.

	WHOQOL-BREF						
	Domain 1	Domain 2	Domain 3	Domain 4	Q1	Q2	Total
AS-WIS-20	−0.407 ***	−0.371 ***	−0.249 **	−0.275 ***	−0.334 ***	−0.280 ***	−0.401 ***
AS-WIS-16	−0.385 ***	−0.359 ***	−0.230 **	−0.268 **	−0.312 ***	−0.251 **	−0.386 ***
MFIS-Total	−0.245 **	−0.207 *	−0.077	−0.121	−0.230 **	−0.189 *	−0.207 *
MFIS-Phy.	−0.307 ***	−0.231 **	−0.061	−0.115	−0.232 **	−0.233 **	−0.224 **
MFIS-Cog.	−0.171 *	−0.183 *	−0.062	−0.102	−0.193 *	−0.136	−0.171 *
MFIS-Psy.	−0.179 *	−0.060	−0.066	−0.136	−0.167 *	−0.111	−0.12–

Domain 1—physical health, Domain 2—psychological health, Domain 3—social relationships, Domain 4—environment, Q1—overall QoL, Q2—general health; AS-WIS-20—work disability risk score on the original scale; AS-WIS-16—work disability risk score on the shortened scale; MFIS-Total—total score for physical fatigue and lack of energy; MFIS-Phy.—physical subscale-MFIS-Cog.—cognitive subscale; MFIS-Psy—psychosocial subscale; * *p* < 0.05, ** *p* < 0.01, *** *p* < 0.001.

**Table 4 ijerph-19-13260-t004:** The general and clinical characteristics of RA patients divided into two groups depending on the level of risk of work disability, *n* (%) or a median (Q1–Q3).

Characteristics	Total *N* = 142		AS-WIS (Score)				*p*-Value
			Low-to-Moderate *N* = 97		High *N* = 45		
	n	%	n	%	n	%	
Gender, male	65	45.8	42	43.3	23	51.1	0.385
Age (years) (Me):	47 (38–58)		45 (37–56)		50 (40–60)		0.115
BMI (kg/m^2^) (Me):	24.9 (22.5–28.6)		24.6 (22.5–27.6)		25.6 (22.8–30.9)		0.194
Marital status, single	28	19.7	25	25.8	3	6.7	0.007 **
Level of education:							0.122
Basic, vocational	25	17.6	14	14.4	11	24.4	
Secondary	43	30.3	27	27.9	16	35.6	
Higher	74	52.1	56	57.7	18	40.0	
Full-time job	110	77.5	79	81.4	31	68.9	0.095
Comorbidities (yes)	51	35.9	33	34.0	18	40.0	0.490
Hypothyroidism	20	14.1	14	14.4	6	13.3	0.933
Hypertension	25	17.6	17	17.5	8	17.8	0.841
Heart disease	11	7.7	6	6.3	5	11.1	0.950
Diabetes	9	6.3	7	7.2	2	4.4	0.719
Asthma	3	2.1	3	3.1	0	0.0	0.052

AS-WIS—work disability risk score, Me—median, **—*p* < 0.01.

**Table 5 ijerph-19-13260-t005:** The clinical characteristics of RA patients divided into two groups depending on the level of risk of work disability.

Medications Used before Biologic Therapy	Total *N* = 142		AS-WIS (Score)				*p*-Value
			Low-to-Moderate (0–14) *N* = 97		High (>14) *N* = 45		
	n	%	n	%	n	%	
NSAID	51	35.9	34	35.1	17	37.8	0.753
Opioids	5	3.5	1	1.0	4	8.9	0.035 *
Glucocorticoids	17	12.0	10	10.3	7	15.6	0.536
Immunosuppressive drugs	14	9.9	8	8.2	6	13.3	0.520
Arechin	13	9.2	10	10.3	3	6.7	0.755
Cyclosporine	4	2.8	3	3.1	1	2.2	1.000
Methotrexate	82	57.8	52	53.6	30	66.7	0.143
Metypred	32	22.5	18	18.6	14	31.1	0.096
Salazopyrin	1	0.7	1	1.0	0	0.0	1.000
Gold salts	2	1.4	2	2.1	0	0.0	1.000
Sulfasalazine	32	22.5	21	21.5	11	24.4	0.011

AS-WIS—work disability risk score, *—*p* < 0.05.

**Table 6 ijerph-19-13260-t006:** The number (percentage) of RA patients in groups differing in terms of the level of risk of work disability and the biologics used, and the results of independence tests.

Biologic Drug	Total *N* = 142		AS-WIS (Score)				*p*-Value
			Low-to-Moderate (0–14) *N* = 97		High (>14) *N* = 45		
	n	%	n	%	n	%	
Adalimumab	13	9.2	9	9.3	4	8.9	1.000
Baricitinib	7	4.9	3	3.1	4	8.9	0.208
Certolizumab pegol	3	2.1	1	1.0	2	4.4	0.236
Etanercept	4	2.8	1	1.0	3	6.7	0.094
Golimumab	14	9.9	11	11.3	3	6.7	0.549
Infliximab	6	4.2	5	5.2	1	2.2	0.665
Rituximab	14	9.9	13	13.4	1	2.2	0.038 *
Secukinumab	42	29.6	29	29.9	13	28.9	0.903
Tocilizumab	39	27.5	25	25.8	14	31.1	0.007

AS-WIS—work disability risk score, *—*p* < 0.05.

**Table 7 ijerph-19-13260-t007:** QoL (WHOQOL-BREF) and fatigue levels (MFIS) in 142 patients with RA divided into two groups depending on the level of risk of work disability (AS-WIS), *n* (%).

	Total *N* = 142	AS-WIS (Score)		*p*-Value
		Low-to-Moderate	High	
WHOQOL-BREF, Me (Q1-Q3)	63 (53–71)	65 (56–73)	58 (50–66)	0.003 **
Physical health	43 (32–54)	47 (36–54)	32 (25–47)	<0.001 ***
Psychological health	71 (54–83)	71 (54–83)	63 (50–75)	0.005 **
Social relations	75 (58–75)	75 (58–75)	67 (58–75)	0.096
Environment	67 (61–74)	67 (61–77)	67 (55–70)	0.146
Overall QoL	4 (3–4)	4 (3–4)	3 (2–4)	0.006 **
General health	2 (2–3)	3 (2–3)	2 (2–3)	0.093
MFIS (0–84 score)	45 (33–56)	44 (34–55)	48 (33–58)	0.492
Physical subscale (0–36)	21 (15–26)	20 (15–25)	22 (15–27)	0.214
Cognitive subscale (0–40)	19 (13–25)	19 (14–25)	18 (13–27)	0.960
Psychosocial Subscale (0–8)	6 (4–8)	6 (4–8)	6 (4–8)	0.060

AS-WIS—work disability risk score; Me—median; MFIS—Modified Fatigue Impact Scale; Q1—lower quartile; Q3—upper quartile, **—*p* < 0.01, ***—*p* < 0.001.

**Table 8 ijerph-19-13260-t008:** The number (percentage) of patients in groups differing in the risk of incapacity for work and selected predictors, as well as estimates of the odds ratios and their 95% confidenc-intervals—one-dimensional analysis.

Predictors	Sickness Absence AS-WIS		*p*-Value	OR (95% CI)
	High	Low/Moderate		
Age > 58 years	19 (42.2%)	22 (22.7%)	0.017 *	2.49 (1.17–5.32)
BMI > 28.6 kg/m^2^	17 (37.8%)	19 (19.6%)	0.020 *	2.49 (1.14–5.46)
Lack of higher education qualifications	27 (60.0%)	41 (42.3%)	0.049 *	2.05 (1.00–4.21)
WHOQOL Total <72	43 (95.6%)	66 (68.0%)	0.001 **	10.1 (2.30–44.4)
WHOQOL Domain 1 <39	28 (62.2%)	29 (29.9%)	<0.001 ***	3.86 (1.84–8.12)
WHOQOL Domain 2 <79	39 (86.7%)	52 (53.6%)	<0.001 ***	5.62 (2.18–14.5)
WHOQOL Domain 3 <75	27 (60.0%)	38 (39.2%)	0.020 *	2.33 (1.13–4.80)
WHOQOL Domain 4 <64	20 (44.4%)	27 (27.8%)	0.050	2.07 (0.99–4.33)
Question 1 (overall QoL) <4	29 (64.4%)	41 (42.3%)	0.014 *	2.48 (1.19–5.14)
Question 2 (general health) <3	28 (62.2%)	47 (48.5%)	0.126	1.75 (0.85–3.61)
MFIS Physical subscale >21	28 (62.2%)	45 (46.4%)	0.079	1.90 (0.92–3.92)
In a relationship	42 (93.3%)	72 (74.2%)	0.015 *	4.86 (1.38–17.1)
Rituximab: no	44 (97.8%)	84 (86.6%)	0.038 *	6.81 (0.86–53.8)
Opioids before biologic therapy	4 (8.9%)	1 (1.0%)	0.035 *	9.37 (1.02–86.4)

Domain 1—physical health, Domain 2—psychological health, Domain 3—social relationships, Domain 4—environment; AS-WIS-20—work disability risk score on the original scale; AS-WIS-16—work disability risk score on the shortened scale; MFIS-Total—total score for physical fatigue and lack of energy; MFIS-Phy.—physical subscale; MFIS-Cog.—cognitive subscale; MFIS-Psy—psychosocial subscale; *—*p* <0.05; **—*p* < 0.01; ***—*p* < 0.001; OR—odds ratio; CI—confidence interval.

## Data Availability

Data is available upon reasonable request. The data can be obtained from the corresponding author upon request after obtaining appropriate ethical approval.

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
