# Peer review of "Work Ability and Quality of Life in Patients with Rheumatoid Arthritis"

_ijerph, 2022, doi:10.3390/ijerph192013260_

Round 1

Reviewer 1 Report

Dear Editor

The comments are in the attached file

Thank you

Author Response

We would like to thank the Editor and the Reviewers for positive evaluations of our paper entitled “Work ability and quality of life in patients with rheumatoid arthritis” with manuscript ID ijerph-1901773.

Below, we address the remaining issues raised in the review. The final version of the manuscript text includes all necessary modifications and improvements as indicated in the attached table.

We are at your disposal for any further modifications that you find advisable.

We very much hope that our carefully prepared response appears comprehensive and proves helpful in obtaining a positive final decision accepting our paper for publication in your prestigious journal International Journal of Environmental Research and Public Health.

With kind regards, awaiting your respected decision,

Wojciech Tański

Reviewer 2 Report

Thank you for giving the opportunity to review this manuscript. While the manuscript is well written, I have few concerns and suggestions that I feel authors should address prior to the manuscript is being considered for publication:

1. The authors adopt a cross-sectional study. I don't think it would be appropriate to assess the QoL for first time patients with rheumatoid arthritis receiving treatment in the hospital. Such attribute should be explored in the context of a prospective study, whereby the QoL may deteriorate or improve over time with the treatment outcome. Hence, a patient diary should have been given at least.

2. It would be more appropriate to sub-section the methods part as study design and setting, study population/sample, inclusion and exclusion criteria, instruments used, procedures, statistical analyses.

3. Authors report the psychometrics properties and internal consistency (for example Cronbach's alpha) based on previous literature. What are those parameters for your current study population/sample, especially for the WHO-QOL brief which is not reported? 

4. Why did the authors only dichotomize the scores, for example the AS-WIS? Forceful dichomization, for example low-moderate will instead reduce the statistical power of analysis.  

5. It would be a little inappropriate to report odds ratios (OR) when you do not have controls (free of the the disease)? Please consult a statistician on the improvement of this aspect. 

Author Response

(The authors gave the same response as above.)

Reviewer 3 Report

In the article entitled: Work ability and quality of life in patients with rheumatoid arthritis, the authors study the relationship between quality of life and the risk of work disability, they also evaluate other factors that negatively impact work disability in RA patients; Although the research question, as well as the objective of the work, are very relevant; how the manuscript is developed is confusing and presents several limitations when was evaluated, I will point out some of them below.

*Major issues

In the introduction: The authors make a very general description of the study background. They need to justify the use of opioids more appropriately and the MFIS and AS-WIS instruments since they refer to the quality of life in patients with RA, but not to the dimensions evaluated by these last two scales. So initially, they do not allow the reader to understand the importance of their use, or why they evaluated the use of opioids.

In Study Material [SIC]: The manuscript does not present an adequate structure for a scientific article; the description of the inclusion criteria and other sections correspond more to a research protocol or a thesis.

In results: The authors do not adequately separate or define a section of results; this is confusing for the reader when interpreting various data. Example: in table 1 the authors refer to the analysis of the reliability of the AS-WIS items of their study sample, but these values ​​should go right into the results section.

There are various grammatical and editorial errors in the different sections of the manuscript.

*Other considerations regarding the manuscript and minor issues

Line 25. The authors do not explain which biological agents they are referring to.

Line 26. The authors indicate in material and methods that they used three instruments; however, they only indicate two in the results: WHOQOL-BREF and AS-WIS, and do not justify the relevance of using the MFIS.

Lines 35 and 36. The authors do not conclude with aspects related to the various instruments used in the study.

Line 39. The authors could use more keywords according to their work.

Lines 51 and 52. The rates of work disability among RA patients are very general; the authors must justify why these values ​​are due; this will allow the reader to have a broader picture of the prevalence of this condition.

Lines 53-56. The authors must delve into the implications of reduced work participation on a global scale and in other contexts, such as mental illness, family dynamics, and the impact on labor and health institutions.

Lines 69-71. The authors need to briefly describe which biological therapies they are referring to and early treatment.

Lines 103 and 104. In the summary, the authors indicate results regarding the use of opioids, but this variable is not contemplated in the hypothesis, so the latter is inadequate or incomplete.

Line 105. The study material part corresponds to the Methodology.

Line 144. The authors had to briefly describe the meaning of the instruments the first time they were pointed out.

Line 151. According to the WHO site (cite 12), the instrument must have a permit; however, is not clear whether such permission was obtained.

The authors use acronyms without clarifying their meaning, for example COPD, Line 330.

Lines 468 and 469. The beginning of the conclusion does not correlate with the most important results of the work.

Author Contributions: The authors did not use acronyms as suggested in the template.

References. Several of the references are incomplete or inappropriately worded, for example: References 2, 5, 10, 12, among others.

Author Response

(The authors gave the same response as above.)

Reviewer 4 Report

The article seems well written and the use of statistics seems appropriate.

Some points of concern are: 1) although the study subjects are described as inpatients in line 24 of the abstract, the Methods section of the text does not specify whether only inpatients or outpatients were included; 2) it appears that the gender of the patients was not mentioned; and 3) the authors may need to discuss the possibility that the causal relationship between quality of life and work disability is bidirectional, especially with regard to psychological health.

As minor points

1) In line 243 of Table 3, ** p<0.001 should be ** p<0.01.

2) The + in line 318 should be omitted.

3) In line 326 of Table 9, the meaning of * and ** should be explained as in the other tables.

Author Response

(The authors gave the same response as above.)

Round 2

Reviewer 2 Report

Thank you for your revisions and justifications. I understand your justifications on the reporting of odds ratios and would agree to your point here for logical interpretations. I am happy with the changes made and I look forward for the paper to appear in the scholarly literature. All the best!

Author Response

Dear Reviewer,

Thank you very much for your positive evaluation of our manuscript. With the suggested corrections, the manuscript is certainly easier for the reader to read and more valuable.

Reviewer 3 Report

The authors have made substantial modifications to their manuscript and have addressed most of the observations made by the evaluator; however, two aspects must be addressed, which are mentioned below:

1) It is recommended that the Inclusion and exclusion criteria section be rewritten as a paragraph and not bulleted.

2) Incorporate the conclusions until the end of the writing

Remove the colon from the conclusions, Limitations of the study, and Practical implications sections.

Author Response

We thank the Reviewer for his comments.

All changes have been highlighted in the revision tracking mode. Inclusion and exclusion criteria for the study have been described as a separate paragraph. In addition, we have moved the results section to the end of the writing. 
We have removed colons from the Conclusions, Limitations of the Study and Practical Implications sections.